

# Investigating the complementarity of thermal and physical soil organic carbon fractions

Amicie A. Delahaie[1], Lauric Cécillon[1], Marija Stojanova[1], Samuel Abiven[1], Pierre Arbelet[2], Dominique Arrouays[3], François Baudin[4], Antonio Bispo[3], Line Boulonne[3], Claire Chenu[5], Jussi Heinonsalo[6], Claudy Jolivet[3], Kristiina Karhu[6], Manuel Martin[3], Lorenza Pacini[1,2], Christopher Poeplau[7], Céline Ratié[3], Pierre Roudier[8], Nicolas P. A. Saby[3], Florence Savignac[4], Pierre Barré[1]

[1] École Normale Supérieure de Paris, Laboratoire de Géologie, France
[2] Greenback (commercial name: Genesis), Paris, France
[3] INRAE, Info&Sols, 45075, Orléans, France
[4] UMR ISTeP 7193, Sorbonne Université, CNRS, Paris, France
[5] UMR ECOSYS, INRAE, AgroParisTech, Université Paris Saclay, 91123 Palaiseau, France
[6] Department of Forest Sciences, Faculty of Agriculture and Forestry, University of Helsinki, Helsinki, Finland
[7] Thünen Institute of Climate-Smart Agriculture, Braunschweig, Germany
[8] Manaaki Whenua - Landcare Research, Palmerston North, New Zealand

*Correspondence to*: Amicie A. Delahaie (amicie.delahaie@ens.fr)

**Abstract.** Partitioning soil organic carbon (SOC) in fractions with different biogeochemical stability is useful to better understand and predict SOC dynamics, and provide information related to soil health. Multiple SOC partition schemes exist but few of them can be implemented on large sample sets and therefore be considered as relevant options for soil monitoring. The well-established particulate- (POC) *vs.* mineral-associated organic carbon (MAOC) physical fractionation scheme is one of them. Introduced more recently, Rock-Eval® thermal analysis coupled with the PARTY$_{SOC}$ machine-learning model can also fractionate SOC into active ($C_a$) and stable SOC ($C_s$). A debate is emerging as to which of these methods should be recommended for soil monitoring. To investigate the complementarity or redundancy of these two fractionation schemes, we compared the quantity and environmental drivers of SOC fractions obtained on an unprecedented dataset from mainland France. About 2,000 topsoil samples were recovered all over the country, presenting contrasting land covers and pedoclimatic characteristics, and analysed. We found that the environmental drivers of the fractions were clearly different, the more stable MAOC and $C_s$ fractions being mainly driven by soil characteristics, whereas land cover and climate had a greater influence on more labile POC and $C_a$ fractions. The stable and labile SOC fractions provided by the two methods strongly differed in quantity (MAOC/$C_s$ = 1.88 ± 0.46 and POC/$C_a$ = 0.36 ± 0.17; n = 843) and drivers, suggesting that they correspond to fractions with different biogeochemical stability. We argue that, at this stage, both methods can be seen as complementary and potentially relevant for soil monitoring. As future developments, we recommend comparing how they relate to indicators of soil health such as nutrient availability or soil structural stability, and how their measurements can improve the accuracy of SOC dynamics models.





## 1 Introduction

Evaluating the biogeochemical stability of soil organic carbon (SOC) is crucial for predicting future SOC stock changes and
assessing soil health. SOC biogeochemical stability depends on many interacting factors such as soil organic matter (SOM)
molecular composition, and interactions with the mineral matrix (von Lützow et al., 2006; Schmidt et al., 2011). For a given
soil, SOC represents a continuum of mean residence times (MRT) ranging from days to millennia (Balesdent, 1996).
However, this continuum cannot be measured directly and then used efficiently for evaluating SOC biogeochemical stability.
For this reason, many studies have proposed SOC fractionation schemes to distinguish fractions with contrasting residence
times, enabling a practical assessment of SOC biogeochemical stability (Poeplau et al., 2018). Nevertheless, many of these
fractionation methods are expensive and time-consuming, making their use on large datasets almost impossible.

Recently, Lavallee et al. (2020) proposed a drastic simplification of the SOC biogeochemical stability continuum by dividing
SOC into two fractions of contrasted stability: particulate (POC) and mineral-associated (MAOC) fractions, following on
early work by Cambardella & Elliott (1992). This physical fractionation scheme is relatively quick, can be implemented on
hundreds of samples (Lugato et al., 2021), and recent studies have underlined the potential interest of such a dualistic view
of the SOC persistence continuum (Cécillon et al., 2021b; Angst et al., 2023; Lugato et al., 2021).

Less popular than physical fractionation, thermal fractionation has also been proposed as an efficient method to evaluate
SOC quality (Plante et al., 2009). In particular, Rock-Eval® thermal analysis has been the subject of a growing interest in
recent years for assessing SOC biogeochemical stability (Saenger et al., 2013; Barré et al., 2016; Sebag et al., 2016;
Soucémarianadin et al., 2018). This method is relatively fast and can be used to analyse a series of thousands of samples
(Delahaie et al., 2023). Moreover, Cécillon et al. (2018; 2021a) developed a machine learning model, PARTY$_{SOC}$, which
uses Rock-Eval® thermal analyses results as input variables to estimate the proportion of SOC that is stable at a centennial
scale and, by difference, the proportion of SOC that is active at this timescale. Kanari et al. (2022) showed that the fractions
determined by PARTY$_{SOC}$ match the "stable" and "active" fractions of the AMG model (Clivot et al., 2019), improving its
simulations of SOC stock evolutions in croplands. As a result, Rock-Eval® thermal analysis associated with PARTY$_{SOC}$
allows partitioning SOC in a more labile fraction ($C_a$) with an MRT ranging from 20 to 40 years, and a stable fraction ($C_s$)
which can be considered inert at a centennial timescale.

The POC/MAOC physical fractionation and the $C_a$/$C_s$ thermal fractionation are therefore two methods that can potentially be
used to split SOC in fractions with contrasted biogeochemical stability and be implemented on large sample sets. With the
growing interest in monitoring programs of soil health and the need for better initialization methods able to improve the
accuracy of SOC dynamics models, it is necessary to assess the extent to which these two fractionation approaches are
complementary or redundant.

Our hypotheses were that as they do not target the same SOC pools (Balesdent, 1996; Poeplau et al., 2018; Kanari et al.,
2022), POC and $C_a$ as well as MAOC and $C_s$ fractions may represent different quantities and have different environmental
drivers (soil characteristics, land cover, and climate variables) and can therefore be considered as complementary. To test our



hypotheses, we used an unprecedented dataset comprising ca. 2,000 Rock-Eval® thermal analyses and ca. 1,000 POC/MAOC physical fractionation data from the analysis of topsoil (0–30 cm) samples that are part of the French soil monitoring network (RMQS).

## 2 Material and methods

### 2.1 RMQS soil samples


The soil samples used in this article are part of the French "Réseau de mesures de la qualité des sols" (RMQS) network and were previously described in Gogé et al. (2012) and Delahaie et al. (2023). A complete description of this monitoring network is available in Jolivet et al. (2006; 2022). Briefly, French mainland soils are monitored every 15 years following a 16 km × 16 km regular square grid, resulting in 2,170 sites. When possible, the sampling site is set at the centre of the cell;

alternatively, another site is selected if needed within a 1 km radius from the centre of the cell. At each sampling site, 25 topsoil samples (0 to 30 cm or tilled layer depth, whichever depth is smaller) are taken from a 20 m × 20 m area using a spiral auger, and then mixed, resulting in a composite sample of 5 to 10 kg.

The composite samples are then placed in trays and air-dried at 30 °C for 8 to 10 days, and quartered according to NF ISO 11464, resulting in a subsample of ca. 650 g. They are then hand-crushed and sieved at 2 mm, and an aliquot is ground under

250 µm by a Cyclotec 1093 (FOSS) (Gogé et al., 2012). The remains of the samples are stored in plastic buckets.

A total of 2,037 samples from the 1ˢᵗ sampling campaign (2000–2009) out of 2170 were recovered and analysed by Rock-Eval® thermal analysis in Delahaie et al. (2023).

### 2.2 Soil and environmental data associated to each RMQS site and topsoil sample

**2.2.1 Soil data**

Physical and chemical analyses were carried out on the composite soil samples at the Laboratoire d'Analyse des Sols (INRAE, Arras, France). The inorganic carbon content ($C_{inorg}$) was derived from the total carbonate content, in grams per kilogram of sample (volumetric method, NF EN ISO 10693), and calculated as $C_{inorg}$ = total carbonate × 0.12; the total carbon content, in grams per kilogram of sample, was determined by elemental analysis using dry combustion on non-

decarbonated soil; the organic carbon content was derived from the elemental analysis (TOCea), in grams per kilogram of sample, and calculated as total carbon content minus inorganic carbon content: TOCea - $C_{inorg}$ (NF ISO 10694 ; "NF" standing for French standard); the total nitrogen, in grams per kilogram of sample, was determined by dry combustion (NF ISO 13878); the particle size distribution was measured without decarbonation, in grams per kilogram of sample (Robinson pipette and underwater sieving, method validated in relation to standard NF X31-107); pH was measured in a suspension of

soil diluted with water (dilution 1 : 5, NF ISO 10390); the exchangeable calcium content, in centimoles per kilogram of sample, was measured by cobaltihexammine chloride extraction (NF X31-130); the exchangeable magnesium content, in centimoles per kilogram of sample, was measured by cobaltihexammine chloride extraction (NF X31-130); the exchangeable



potassium content, in centimoles per kilogram of sample, was measured by cobaltihexammine chloride extraction (NF X31-130); the free iron oxides, in grams per 100 g, measured with the Tamm method in the dark (amorphous oxides) and Mehra–Jackson method (crystalline oxides) (INRA standard/NF ISO 22036).

### 2.2.2 Climate data

We allocated to each sampling site the climatic data corresponding to the SAFRAN 8 km x 8 km grid-cell based on where the cell was located (https://publitheque.meteo.fr /okapi/accueil/okapiWebPubli/index.jsp, last access: 31 March 2022). The daily data were averaged over the 1969–1999 period (i.e. the 30-year common period before the first sampling campaign starting in 2000) in order to compute the mean annual temperature (MAT) and mean annual precipitation (MAP) for each site.

### 2.2.3 Land cover data

Land cover data were recorded during sampling. Four main categories of land cover were considered for this study: "croplands", "forests", "grasslands", and "vineyards & orchards". A few samples were collected in "wastelands", "urban parks", and "sites with little human disturbance". Considering the very small number of samples, "wastelands" (ca. 10) and "gardens" (n = 3) were not included in this study. The number of samples from environments with little human disturbance (ca. 30) could potentially be considered sufficient for statistical treatment; however, these samples represent a very heterogeneous set (10 miscellaneous subclasses, such as peatlands, alpine grasslands, water edge vegetation, heath, and dry siliceous meadows). Thus, those sites were also discarded.

## 2.3 Thermal SOC fractionation

### 2.3.1 Rock-Eval® thermal analyses

In total, 2,037 samples were analysed by Rock-Eval® thermal analysis (Disnar et al., 2003; Baudin et al., 2015). For each sample, ca. 60 mg of finely ground matter (< 250 µm) was placed in a special high-temperature-resistant stainless-steel pod, allowing the transport gas to pass through, and then placed inside a Rock-Eval® 6 (RE6) Turbo device (Vinci Technologies). There, it underwent a first phase of pyrolysis under an inert atmosphere ($N_2$) from ambient temperature to 650 °C (three-minute isotherm at 200°C and then a temperature ramp of 30°C $min^{-1}$), and a second phase of oxidation under the laboratory atmosphere purged from water and $CO_2$, from 300 °C to 850 °C (one-minute isotherm at 300°C and then a temperature ramp of 20°C $min^{-1}$). During the pyrolysis phase, hydrocarbon effluents were monitored by a flame ionisation detector, and CO and $CO_2$ were monitored by infrared detectors. During the oxidation phase, CO and $CO_2$ were monitored by infrared detectors. The resulting thermograms were processed using the Geoworks software (Geoworks V1.6R2, Vinci Technologies, 2021).

The organic carbon yield was defined as the ratio of the total organic carbon amount measured by Rock-Eval® thermal analysis (TOCre6, calculated from thermogram area integration) over the total organic carbon amount measured by elemental analysis (TOCea). We chose to apply a quality criterion on this yield: further study was conducted only on



samples with an organic carbon yield ranging from 0.7 to 1.3. This range was set to identify the acceptable yields ensuring the quality of the Rock-Eval® analysis as well as the identity of the sample. Of the 2,037 samples analysed by Rock-Eval® thermal analysis, 1,891 presented an organic carbon yield ranging from 0.7 to 1.3 (Delahaie et al., 2023). We also removed

12 samples with TOCea > 120 g kg$^{-1}$ to avoid organic soils (Eggleston et al., 2006), resulting in a dataset of 1,879 samples.

### 2.3.2 The PARTY$_{SOC}$ fractionation

The PARTY$_{SOC}$ model (Cécillon et al., 2018; 2021a) is a machine-learning model using the results of the Rock-Eval® thermal analysis of mineral topsoils as entry variables. This model, trained on data from long-term agronomic experiments,

uses 18 Rock-Eval® parameters as input variables and determines the proportion of the centennially-persistent organic carbon pool in a mineral topsoil sample. The stable SOC proportion is now calculated routinely by the Geoworks software (Geoworks V1.6R2, Vinci Technologies, 2021) using the model PARTY$_{SOC}$ v2.0EU published in Cécillon et al. (2021a). For each site, we multiply the proportion of stable C by the TOCea to calculate the $C_s$ pool (g C kg$^{-1}$ sample); the active pool $C_a$ (g C kg$^{-1}$ sample) is obtained by difference: $C_a$ = TOCea - $C_s$.


### 2.4 Physical SOC fractionation

The physical SOC fractionation, i.e. particle size fractionation was conducted by the SADEF laboratory (Aspach-Le-Bas, France) on a subset of the RMQS (997 sites) following a protocol based on the norm NF X 31-516, itself based on Balesdent et al. (1991; 1998). The dispersion was carried out with a solution of sodium hexametaphosphate at a concentration of 5 g L$^{-1}$.

$^{1}$. 50 g of soil sieved at 2 mm was stirred in 180 ml of hexametaphosphate solution with 10 glass beads of 5 mm diameter and underwent rotary agitation for 16 hours at 20 °C at 45 rpm in a 250 mL flask. The fractions were then sieved by hand at 0.2 mm with rotative movements, and sprays of demineralised water were used to complete the sieving process. The matter remaining on the sieve was transferred in a capsule, dried, and crushed. The suspension containing the fine particles (< 0.2 mm) and rinsing water were collected for further sieving to 0.05 mm. The same principle was then applied for the sieving at

0.05 mm, but it was conducted on three or four successive fractions of the suspension to avoid clogging the sieve. The liquid fraction containing the particles below 0.05 mm was recovered in a 1 L crystallizer and dried. The drying of the fractions was carried out in a ventilated oven at 105 °C.

This fractionation process thus resulted in three fractions: the mineral-associated organic matter fraction (MAOM) corresponds to the 0–50 μm fraction, the fine particulate organic matter fraction (fPOM) corresponds to the 50–200 μm

fraction, and the coarse particulate organic matter fraction (cPOM) corresponds to the 200–2000 μm fraction. The carbon contained in the MAOM fraction constitutes the MAOC while the carbon contained in both the fPOM and cPOM constitutes the POC.

After drying, all the dry matter in each fraction was recovered. Each fraction was introduced into a corundum bowl and ground with corundum balls (Retch PM400 planetary ball mill) at 400 rpm for 5 minutes to ensure the final matter is ground

at < 250 μm and homogenised.




Carbon and nitrogen measurements were carried out on a Flash 2000 Elemental Analyzer for soils without carbonates (determined by acid test) following the norms NF ISO 10694 and NF ISO 13878, respectively. For carbonated soils, only nitrogen was measured on the Flash 2000 Elemental Analyzer (Dumas method NF ISO 13878). Organic carbon was analysed by chemical oxidation (NF ISO 14235). The total organic carbon retrieved after physical fractionation is noted

TOCfr and the organic carbon yield for this fractionation was defined as the ratio of TOCfr over TOCea.

The C yield was on average 93.4% for the 997 samples. For the same reasons as above, we also introduced a quality criterion on this yield, identical to the one for the Rock-Eval® results (0.7 to 1.3). Following this rule, 33 samples were removed from the dataset. Then, four samples with TOCea > 120 g kg$^{-1}$ were also removed to avoid organic soils (Eggleston et al., 2006), resulting in a final dataset comprising 960 fractionation results. The removed samples showed no particular pedoclimatic

characteristics (Fig. A1, Appendix).

As the physical and thermal fractionations were not conducted on all the samples, there are samples for which data of only one method was available. The intersection of the physical fractionation dataset and thermal fractionation dataset consists of 843 samples and is thereafter designated as the "intersection dataset".


## 2.5 Statistical analysis

### 2.5.1 The determination of drivers of the POC, MAOC, $C_a$, $C_s$, and TOCea quantities

In this study, we tested the influence of different environmental variables on the $C_s$, $C_a$, MAOC, and POC content by using Random Forest regression models based on the method used by Georgiou et al. (2022). One advantage of Random Forest

models is that they can cope with non-normally distributed and correlated variables (Breiman, 2001). The considered environmental drivers were related to soil characteristics (particle size distribution, pH, inorganic carbon content, exchangeable cations contents (calcium, magnesium, potassium), amorphous and crystalline iron oxyhydroxides contents), climate (mean annual precipitation, mean annual temperature), and land cover. The relative importance of each of these features as estimated by the Random Forest model allowed us to evaluate the main drivers of the quantity of each fraction.

The modelling pipeline was divided into two steps. The first preprocessing step used one-hot encoding (creating one boolean column per class for any categorical variable), while all the numeric variables were standardised by removing their mean and dividing them by their variances. In the second step, a bootstrapped Random Forest regressor was used to calibrate the actual model. Grid search and cross-validation were used to choose the model's hyper-parameters (Table A1, Appendix), using cross-validated $R^2$ (determination coefficient) as a performance metric. These hyper-parameters define the structure and

construction of the forest's trees, and as such they are crucial and must be chosen wisely to ensure that the model does not overfit or underfit the training data.

After the model was calibrated, the importance of the environmental predictors was calculated using two different methods: the mean decrease in impurity (MDI) and the Permutation Importance (PI) score (Louppe, 2014). Both methods gave a measurement of the importance of the environmental variables selected in the model. MDI aims at selecting the predictors



that, on average, produce trees with the purest nodes and leaves. In this case, purity refers to the similarity of the samples contained in a single leaf. MDI is calculated using the training set and it is the default decision metric used in constructing scikit-learn's random forests. However, the MDI has a known tendency to lower the importance of the low-cardinality variables and possibly has a bias towards highly-correlated variables (Louppe et al. 2013). The PI score measures the impact of each individual predictor by randomly permuting it and then calculating the increase in prediction error as opposed to

using the non-permuted predictor. By construction, PI can be calculated on both the training and the test set. Its main advantage over MDI is that it shows no bias towards high-cardinality predictors. Highly-correlated variables can be detected using PI as their permutation will have little to no impact on the model's prediction accuracy.

In order to analyse the variable importance results, the environmental variables were grouped into three broad categories: "land cover", "pedology" (particle size distribution, pH, inorganic carbon content, exchangeable calcium content,

exchangeable magnesium content, exchangeable potassium content, amorphous, and crystalline iron oxyhydroxides contents), and "climate" (mean annual temperature and mean annual precipitation). These groupings were used to guide visual analysis of the variable importance plots generated for the $C_s$, $C_a$, MAOC, and POC models.

The Random Forest modelling (and associated metrics) was done in Python as implemented in the the scikit-learn v1.3.0 library (Pedregosa et al, 2011), while the least-squares linear regression used the scipy v1.10.1 library (Virtanen et al., 2020).


### 2.5.2 Assessments of the effect of land-cover on fractions

To assess the effect of land cover on the different fractions, we performed pairwise comparisons of medians using non-parametric Kruskal–Wallis tests ($p < 0.05$) followed by Wilcoxon tests, with $p < 0.05$ for each pair. The correction of p values within the framework of the multiple comparisons was done using the Holm–Bonferroni method. Correlations

between parameters were calculated using the Spearman method.

The data processing and statistical analysis were carried out using R software (V4.1.2; R Core Team 2021): the corrplot (Wei and Simko, 2021), car (Fox and Weisberg, 2019), ggplot2 (Wickham, 2016), ggpubr (Kassambara, 2023a), factoextra (Kassambara and Mundt, 2020), plot3D (Soetaert, 2021), and rstatix (Kassambara, 2023b) packages were added.

## 3 Results

### 3.1 POC *vs.* $C_a$, MAOC *vs.* $C_s$: fractions in different quantities

Figure 1 shows the quantities of $C_s$ plotted against MAOC and the quantities of $C_a$ plotted against POC, comparing two by two the more stable and more labile fractions for each fractionation scheme for the intersection dataset (samples having been subjected to both thermal and physical fractionation schemes). Regarding the more stable fractions (Fig. 1a), the MAOC

content was much higher than the $C_s$ content (on average 19.13 g kg$^{-1}$ of sample versus 10.06 g kg$^{-1}$ of sample for the 843 samples of the intersection dataset). The average ratio of MAOC/$C_s$ for the samples of the intersection dataset was 1.88 ± 0.46. For the more labile fractions (Fig. 1b), the POC content was much lower than the $C_a$ content (on average 5.32 g kg$^{-1}$ of sample versus 14.40 g kg$^{-1}$ of sample for the 843 samples of the intersection dataset). The average ratio of POC/$C_a$ for the



samples of the intersection dataset was $0.36 \pm 0.17$. The correlations between $C_s$ and MAOC on one side (0.90) and between

$C_a$ and POC on the other side (0.87) are both very significant. In comparison, the correlation coefficients between $C_s$, MAOC, $C_a$ and POC on the one hand, and TOC on the other hand are 0.91, 0.96, 0.98, and 0.86 respectively.

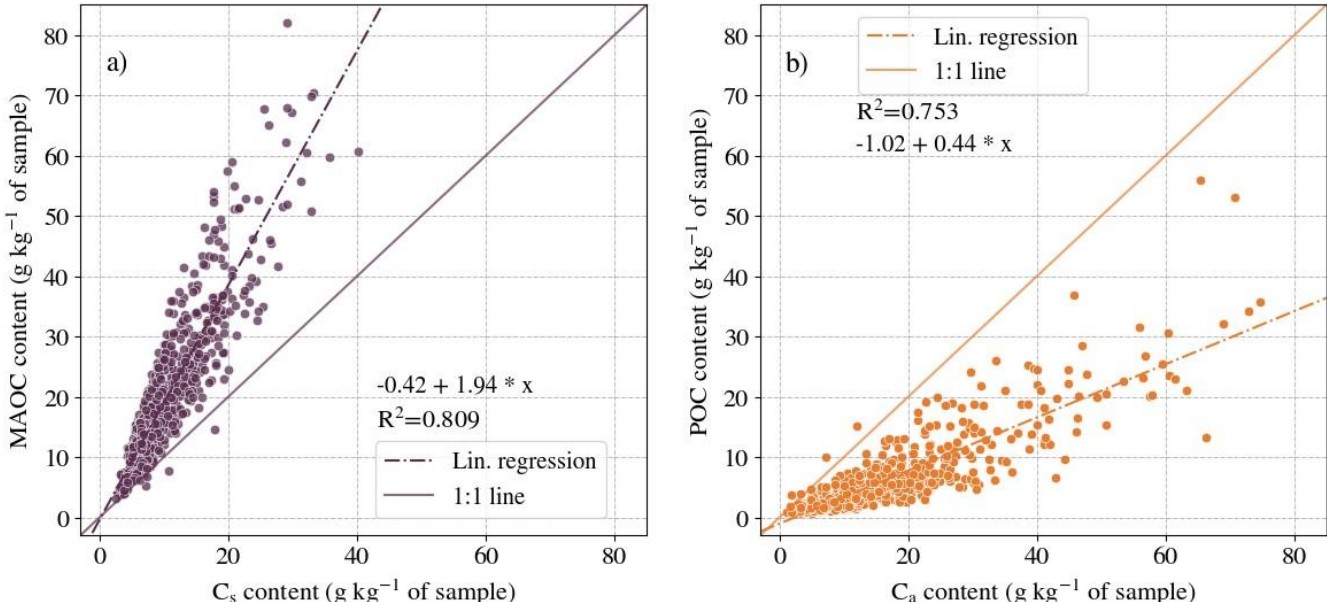

**Figure 1: Comparison of the quantities of the more stable and more labile fractions for the physical and thermal SOC fractionation schemes, with their correlation coefficient $R^2$ and linear regression. Panel (a) shows the quantities of MAOC plotted against $C_s$. Panel (b) shows the quantities of POC plotted against $C_a$. The dataset is the intersection dataset, i.e. samples for which thermal and physical data are available (n = 843).**

**3.2 Differences in SOC fractions' proportions under different land covers**

Figure 2 shows $C_a$ and POC as a proportion of TOC for the four considered land covers. As $C_a+C_s$ and POC+MAOC were equal to TOC, the analysis on $C_s$ or MAOC proportions would have given the same information. Figure 2 shows that the proportion of $C_a$ of the TOC followed the order vineyards & orchards < croplands < grasslands < forests (and similarly for POC). The mean values of the proportion of $C_a$ were 0.48, 0.60, and 0.62 in croplands, grasslands and forests, respectively. It

also shows that POC as a proportion of TOC was significantly smaller in croplands compared to forests, grasslands, and vineyards & orchards, but the median value of POC in croplands was close to the median value in grasslands (0.13 in croplands, 0.17 in grasslands, 0.27 in forests, and 0.19 in vineyards & orchards). As for the median values, the mean values showed a similar difference between croplands ($0.15 \pm 0.05$) and grasslands ($0.19 \pm 0.07$), with forests displaying a higher value ($0.28 \pm 0.09$). While the $C_a$ fraction generally represents a high proportion of TOC (up to 0.75 in forests; proportion of



$C_a$ > 0.5 in 1277 out of 1879 samples), POC most often represented only a minority of TOC (proportion of POC > 0.5 in 17 out of 960 samples).

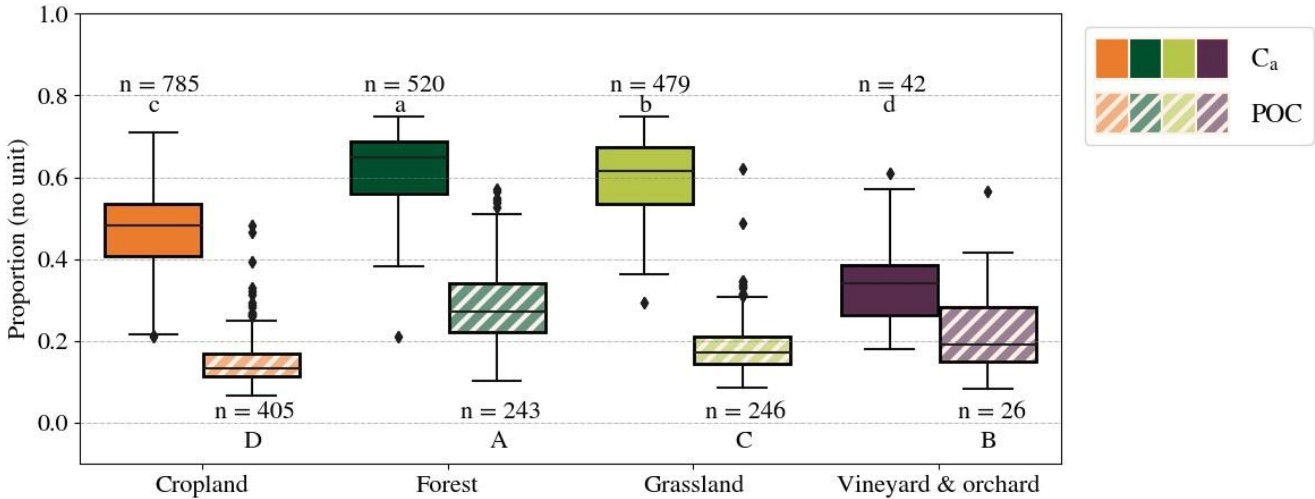

**Figure 2: Proportion of the $C_a$ and POC fractions depending on the land cover. The black line in each box is the median, the lower and upper edges of the black rectangle are the respective first (Q1) and third (Q3) quartiles, and**
**the lower and upper whiskers are the maximum between the minimum value or the first quartile minus 1.5 times the interquartile range (max [min; $Q_1$-1,5×($Q_3$-$Q_1$)]) and the minimum between the maximum or the third quartile plus 1.5 times the interquartile range (min [max; $Q_3$+1,5×($Q_3$-$Q_1$)]), respectively. Different letters indicate significant differences in the distribution of the values for the land covers according to a Kruskal–Wallis test (p < 0.05) and a pairwise Wilcoxon rank sum test (p < 0.05); lowercase letters are used for $C_a$ and uppercase for POC.**


Additionally, different ratios (MAOC/$C_s$, POC/$C_a$ and (MAOC/$C_s$)/(POC/$C_a$) are given in Figure A2, Appendix. Croplands and grasslands exhibited a similar and small POC/$C_a$ ratio, while forests and vineyards & orchards show a higher POC/$C_a$ ratio. On the contrary, the MAOC/$C_s$ ratio is very small in the vineyards & orchards, and the highest in grasslands, with a significant difference between croplands and grasslands.


### 3.3 Drivers of the different SOC fractions quantities

The Random Forest models fitted for the four different SOC fractions aimed at elucidating the relative importance of the soil and environmental variables. Their explanatory capacity was evaluated based on the $R^2$ values obtained for each fraction. The Random Forest models' $R^2$ scores on the test set were 61 % for $C_s$, 53 % for MAOC, 57 % for $C_a$, 36 % for POC, and 58
% for TOCea (Fig. A2, Appendix). The $R^2$ values obtained on cross-validation for the train set were higher but close (Table



A1, Appendix), except for the POC which has a strongly higher train set score. Overall, the models' performance was satisfactory, and allowed their variable importance to be analysed.

Figure 3 shows the importance of the different categories of environmental variables on the quantities of POC, $C_a$, MAOC, and $C_s$, evaluated using two different methods. The results with both the MDI and the PI showed a higher importance of the

pedological features and a smaller importance of climate and land cover in the more stable fractions ($C_s$ and MAOC). On the contrary, the more labile fractions ($C_a$ and POC) showed a higher importance of land cover and, to a lesser extent, climate, compared to $C_s$ and MAOC. $C_a$ tended to show a slightly stronger influence of climate and weaker influence of land cover, while POC rather showed the opposite. Among the more stable fractions, climate and land cover tended to have a slightly higher importance for MAOC than for $C_s$. The results for the TOCea showed a mixture of drivers, in between stable and

labile fractions.

The results for the Spearman correlation coefficients between all environmental variables and fractions quantities are given in Appendix in Table A2. Soil variables favouring organic matter/mineral interactions (clay, metallic oxides, CEC, exchangeable calcium) were positively correlated with fractions. Overall, these correlations were stronger for the $C_s$ and MAOC fractions. On average, iron oxyhydroxides and exchangeable cations are the most important factors influencing the

size of the fractions (Figure 3). Carbonates and pH little influenced the size of the fractions and texture had a minor role but for $C_s$. Regarding climate variables, MAT had a higher influence than MAP except for $C_s$ fractions.

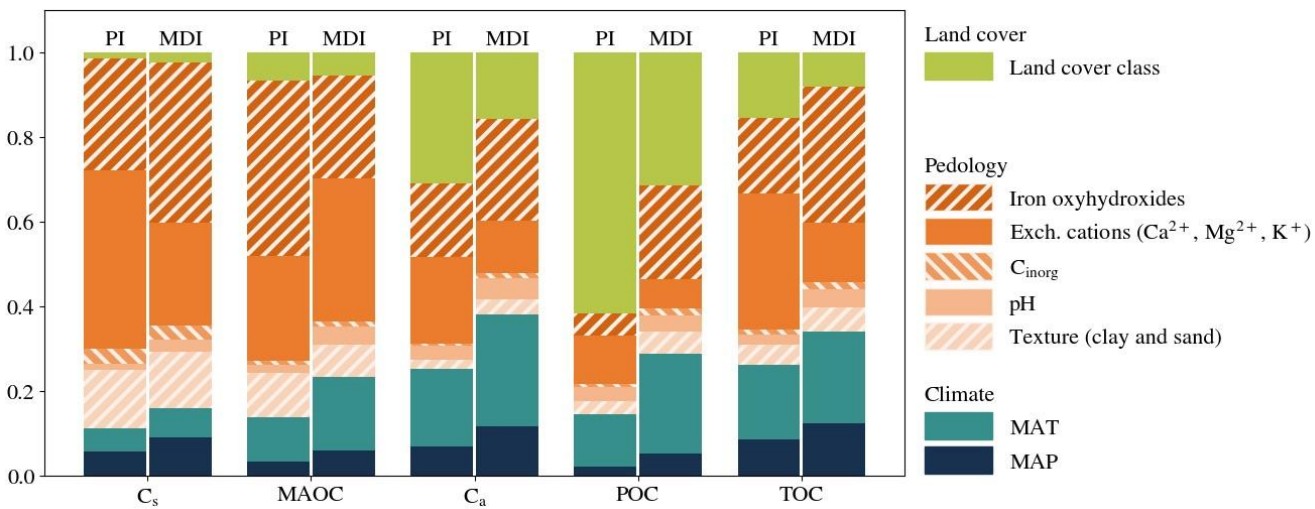

**Figure 3: Importance of the different categories of soil and environmental variables (climate, pedology, and land**
**cover) for the four fractions $C_s$, MAOC, $C_a$, and POC, and TOCea as a comparison (in g C kg$^{-1}$ sample), assessed**
**using MDI and PI.**



## 4 Discussion

### 4.1 A strong influence of land cover on the more labile SOC fractions


This study, based on an unprecedented number of measurements (n = 960 for POC/MAOC and n = 1,879 for $C_a/C_s$), shows the high influence of land cover on the relative quantity of SOC labile and stable fractions (Fig. 2).

The higher proportion of POC in forest than in grassland and cropland soils (0.13, 0.17, and 0.27 in croplands, grasslands, and forests, respectively) was also observed in Lugato et al. (2021) who used results from 352 physical fractionations
obtained using a fractionation protocol similar to ours on samples from the LUCAS soil monitoring network. It was also observed by Hansen et al. (2024) on a global dataset combining physical fractions obtained with different protocols. Other studies showed similar results for POC, but comparisons are less straightforward as the physical fractionation protocols used are significantly different from those used in our study. For instance, some studies used the protocol of Zimmerman et al. (2007), where the sorting size for POC is > 63 µm (compared with > 50 µm in our protocol) and part of POC is also
recovered after the disruption of sand-sized aggregates. When combining free POC and occluded POC (which corresponds to what is called POC in our study), Poeplau & Don (2013) found proportions of POC of 0.15 in croplands, 0.21 in grasslands, and 0.27 in forests in a variety of sites across Europe which is very close to the values observed here for French soils. The value of 0.13 for the proportion of POC in croplands is also close to the 0.15 value used in Angers et al. (2011) to estimate the proportion of POC from TOC. Chen et al. (2019) gathered data from multiple previous studies to derive POC
proportion values of 0.15 for croplands, 0.31 for grasslands and 0.34 for forests.

Regarding the $C_a$ fraction, we found a higher proportion of $C_a$ in forest than in grassland and cropland soils (0.48, 0.62, and 0.65 in croplands, grasslands, and forests, respectively). This was also observed in previous works, but the body of literature available for discussion is much more limited than for physical fractionation. We can note that the median value of $C_a$ for croplands is close to the mean value (0.48 vs 0.42) obtained by Kanari et al. (2022) on nine long-term field experiments in
mainland France. Moreover, the proportion of $C_a$ is set to 0.60 in the AMG model for grasslands (Clivot et al., 2019) in its default parameterisation which is also in line with the $C_a$ proportions observed for grassland sites of the RMQS (0.62 on average). The difference is bigger regarding croplands as the default parameterisation for $C_a$ is set to 0.35 while the value found for the RMQS is 0.48. This may be explained by the fact that the AMG model was developed on long-term experimental sites that have been cropped for several decades, whereas French agricultural soils have probably on average
undergone more changes in vegetation cover. The higher $C_a$ value in RMQS cropland soils could thus be due to a difference in land-cover history in French cropland soils compared to cropland sites of long-term experiments.

The similar proportions of POC in vineyards & orchards and grasslands were less expected as C inputs and contents are reduced in vineyards & orchards (grass cover was very sparse in vineyards & orchards at the time of the sampling). This may be explained by differences in the composition of the particulate organic matter for the different land covers, as the
particulate organic matter groups together particles that can be biochemically quite heterogeneous (Schrumpf et al., 2013; Soucémarianadin et al., 2019). In our study, the POC fraction in vineyards & orchards topsoils might be much more





biogeochemically stable than the same fraction in cropland and grassland topsoils because of the presence of more lignified woody debris and pyrogenic C derived from the combustion of vine shoots. An extreme example of soil with high proportion of POC and $C_s$ can be found at the Versailles long-term bare fallow site where topsoils have a proportion of $C_s$ close to 1 and

a POC proportion around 0.30, constituted essentially of charcoal (Chassé et al., 2021). Our results therefore suggest that POC/MAOC fractionation gives an erroneous view of biogeochemical stability for vineyards & orchards, which have the same POC proportion as grassland but are very C depleted (9.5 vs 24 g C kg-1). The biochemical nature of the fractions is unaccounted for in our study as this data is not available, but it is probably an important driver for POC quantities. The fact that such an important driver is not taken into account may explain the poorer performance of the model for POC compared

to the other fractions (36% vs 55-60% for the other fractions).

### 4.2 SOC fractions with different quantities, drivers and biogeochemical stabilities

The significant differences between the $C_s$ and MAOC quantities on one side, and $C_a$ and POC quantities on the other side (Fig. 1), show that they do not correspond to the same fractions. This result was expected due to the definition of the four

fractions, but it is evidenced for the first time on a large dataset. Indeed, previous studies using isotopic measurements observed that the mean residence times for MAOC ranged from decades to centuries (Anderson & Paul, 1984; Balesdent et al. 1987; Balesdent 1996; von Lützow et al., 2007; Kleber et al., 2015). By definition, $C_s$ corresponds to centennially stable SOC (Cécillon et al., 2018; 2021a). This implies that MAOC encompasses also a certain amount of relatively labile SOC and therefore explains its larger quantity compared to $C_s$. Potentially, this could partly be related to the fractionation method

itself, in which only size fractionation is employed to separate fractions, so that the MAOC fraction might also contain a certain amount of fine POC or soluble compounds (Lavallee et al., 2020; Cotrufo et al., 2023). Conversely, the $C_a$ fraction which corresponds to SOC fraction with a mean residence time of 20-40 years (Kanari et al., 2022) is larger than the POC fractions dominated by SOC with a generally shorter MRT (Balesdent, 1996; von Lützow et al., 2007).

This ranking in terms of MRT, $C_s$ > MAOC > $C_a$ > POC, is also in line with the different environmental drivers explaining

the quantities of these four fractions, whatever the selected method (MDI or permutation) (Fig. 3). Indeed, we observed that the importance of land cover is much higher for POC and $C_a$ than it is for $C_s$ and MAOC, whereas pedological variables are much more important for $C_s$ and MAOC fractions. The importance of land cover for SOC with lower MRT has already been documented in several studies. For instance, Poeplau and Don (2013) observed that POC fractions are very sensitive to land cover in topsoils and Balesdent et al. (2018) showed that land cover is a major driver of the incorporation of "young" C in

topsoils indicating on average a smaller portion of SOC with low MRT in croplands compared to grassland and forest topsoils. The fact that SOC with higher MRT is mostly driven by soil variables and that SOC with lower MRT is mainly explained by land cover and climate was also evidenced by Mathieu et al. (2015) using [14]C in 122 profiles of mineral soil across the world. They observed that the age of topsoil organic carbon, which is on average less biogeochemically stable than deep SOC, was primarily affected by climate and land cover whereas, the age of deep soil carbon was affected more by

soil type and soil characteristics such as clay content and mineralogy. The little influence of land cover on the stable





fractions was somehow expected as most of French soils have undergone a series of land cover changes during the last millenia during which the stable fraction formed. Regarding the bulk SOC, our results are in line with those of Edlinger et al. (2023), who used a similar methodology on a smaller dataset to investigate the drivers of SOC. While their features were not strictly identical to ours (no iron oxyhydroxides for instance, but more climatic features), the main categories that stand out

as drivers of the SOC are pedology (mostly exchangeable calcium) and climate, which is what we observed.

Among pedological variables, iron oxyhydroxides (mostly crystalline oxides) and exchangeable cations (mainly calcium cations) were the factors with the greatest influence on the size of the most stable $C_s$ and MAOC fractions. The strong influence of exchangeable cations and oxides on MAOC has also been recently documented in a study involving 16 agricultural sites in the United States (King et al., 2023). The influence of iron oxides and hydroxides on SOC

biogeochemical stability is a well-known fact (Kögel-Knabner et al., 2008), although it was pointed out that crystalline oxides were less efficient at providing bonding sites for SOM. The importance of exchangeable cations, notably calcium, on SOC biogeochemical stability was previously documented (Rowley et al., 2018; 2021). Indeed, calcium cations can strengthen the interactions between 2:1 clay minerals and SOM, both negatively charged, or enable the formation of co-precipitates with SOM.

Overall, our study confirms results from recent studies conducted at regional and global scales showing that physical fractions have different drivers (King et al., 2023 ; Hansen et al., 2024) and strongly supports the idea that it is relevant and informative to consider SOC fractions (either physical or thermal) instead of TOC alone. However, it is difficult to compare our results directly with those of these recent studies. Indeed, the data processing strategies are different and the explanatory variables considered are not the same. Notably, we observed that land cover and soil variables such as exchangeable calcium

and iron oxides were explanatory variables of primary importance in explaining the quantities of POC and MAOC respectively. These variables were not considered as explanatory variables in the path analyses developed by Hansen et al. (2024) which likely explains the low explained variations provided by these path analyses. Conversely, we did not consider NPP as explanatory variable which was found to be a significant driver of MAOC quantities by Hansen et al. (2024). We therefore consider that new data that will probably arrive in the next few years will enable us to refine the drivers of physical

and thermal fractions at different spatial scales.

**4.3 Which fractionation method to use to assess SOC biogeochemical stability in soil monitoring networks?**

Several recent high-level political initiatives have highlighted the importance of soil health for food security and climate change mitigation and adaptation. These include, for instance, the UNFCCC Koronivia joint work action, the 4p1000

initiative ([www.4p1000.org](www.4p1000.org), Rumpel et al., 2020) and the new "Soil Monitoring Law" proposed by the EU. These initiatives emphasize the importance of SOC by highlighting the C sequestration potential of soils, or by stressing the strong influence of SOC on soil health. This general political context is favourable to the development and support of soil monitoring networks. In these networks, SOC content is always measured. While this data is important, information on SOC biogeochemical stability would be particularly useful. Indeed, most soil functions related to SOM, such as nitrogen



mineralization, actually depend on its decay (Janzen, 2006) and assessing biogeochemical stability is also of primary importance to simulate SOC stock evolution (Luo et al., 2016). In this context, the development of indicators of SOC biogeochemical stability that can be implemented on large sample sets is of particular relevance.

Both physical and thermal fractionation methods are good candidates for this. Indeed, they both split SOC in two fractions of different biogeochemical stability using protocols that can be applied to large sample sets. Each has its advantages and

drawbacks regarding its large-scale implementation for soil monitoring. The thermal method is faster (one hour per sample), highly reproducible (Pacini et al., 2023) and, at least in France, less costly than the physical method (< 50€ per sample vs > 100€ per sample in commercial laboratories). POC/MAOC fractionation, on the other hand, requires no expensive equipment, and is already used worldwide. Moreover some studies have proposed to predict POC/MAOC fractions using a machine learning model to make the method faster (Cotrufo et al., 2019; Lugato et al., 2021). However, a recent study

showed that the results of such prediction methods can be questionable and even misleading (Begill et al., 2023).

In this context, the question arises as to what method should be used to determine biogeochemical stability in soil monitoring networks. Our study, based on an unprecedented sample set, reveals that the POC vs $C_a$ and MAOC vs $C_s$ fractions are significantly different in size and do not have exactly the same environmental drivers, meaning that they are not biogeochemically equivalent. This suggests that the two fractionation methods provide, at least partly, different information,

and could be, at that stage, be seen as complementary. Furthermore, they can also be used to answer different questions: For example, physical fractionation methods can be used in combination with isotopic measurements (e.g. $^{13}$C), to study transformation and stabilisation processes of organic matter in soils (Cotrufo et al., 2015). Moreover, it is probably premature to assess the relevance of the two protocols at this stage, as interesting data will be provided by the monitoring networks over the next few years. For instance, with new SOC stock measurement campaigns, it will be possible to have

measurements of SOC stock evolution allowing the proper evaluation of model SOC stock projections at network scales (Le Noë et al., 2023). In addition, several indicators of soil functions are to be measured at large scale for soil health assessment. All these new data will enable us to assess the extent to which the information on SOC biogeochemical stability provided by fractionation results can be used to improve the accuracy of SOC stock evolution simulations, and to gain a better understanding of soil functioning. Such upcoming studies are likely to bring new key elements to the emerging question of

the redundancy or complementarity of physical and thermal fractionation schemes.

**5 Conclusion**

This study allowed us to compare the POC/MAOC physical fractionation and thermal fractionation on an unprecedented amount of samples with an interesting diversity with respect to pedological characteristics, climatic characteristics and land

covers. We showed that both the stable ($C_s$ and MAOC) and labile ($C_a$ and POC) fractions strongly differ in quantities. While the environmental drivers were close for the two stable fractions (respectively the two labile fractions) with a predominance of the soil characteristics (respectively the climate and land cover), they still presented differences suggesting that $C_s$ and MAOC (respectively $C_a$ and POC) correspond to different fractions with different biogeochemical stability. This



means that both fractionation techniques display different thus complementary information. Future work will enable us to

discuss the relevance of one technique rather than the other on a case-by-case basis, depending on the soil properties studied.

## Code availability

The codes for the Random forest and the plots are available on Zenodo:

Stojanova, M., & Delahaie, A. (2024). SOC fractions drivers. Zenodo. https://doi.org/10.5281/zenodo.10551240.

## Data availability

Data on basic soil properties are freely available from the GIS Sol dataverse website: https://data.inrae.fr/dataset.xhtml?persistentId=doi:10.15454/BNCXYB.

## Author contributions

Amicie Delahaie, Lauric Cécillon, Pierre Barré, François Baudin and Claire Chenu designed the study. Florence Savignac and François Baudin produced the Rock-Eval® thermal analyses. Dominique Arrouays, Antonio Bispo, Line Boulonne,

Claudy Jolivet, Manuel Martin, Céline Ratié and Nicolas Saby provided the detailed pedoclimatic data. Amicie Delahaie processed and interpreted the data with the contribution of all co-authors. Marija Stojanova wrote the Random Forest algorithm and provided technical support for the use of the code. Amicie Delahaie and Pierre Barré wrote the manuscript with contribution of all the co-authors.

## Competing interests

Some authors are members of the editorial board of SOIL.

## Acknowledgements

The École Normale Supérieure of Paris is greatly acknowledged for the funding of the PhD thesis grant of Amicie Delahaie. The ADEME (Rock-Eval®-RMQS project, convention n°2003C0017) is acknowledged for their support. Pierre Roudier is funded by the New Zealand Government to support the objectives of the Global Research Alliance on Agricultural

Greenhouse Gases. The authors thank the FREACS project funded by the external call of the EJP Soil (ANR-22-SOIL-0001). We thank Sophie Cornu for helping us to recover the RMQS sample collection.



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



## Appendices

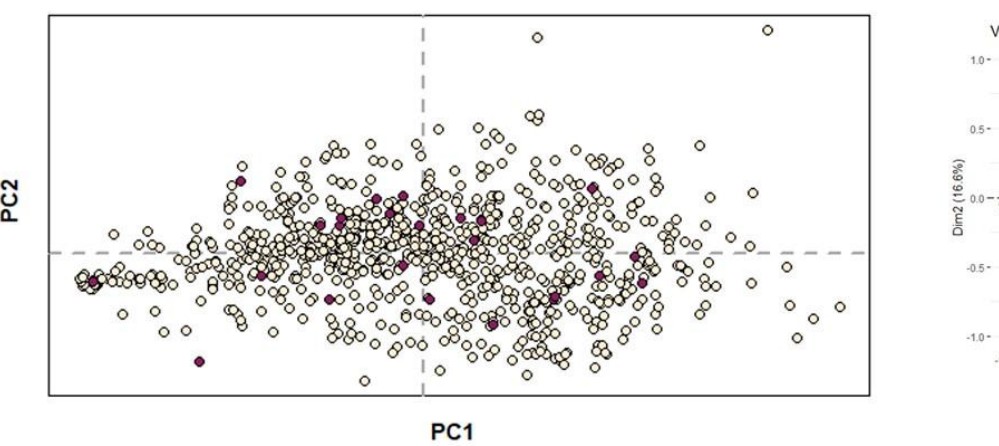

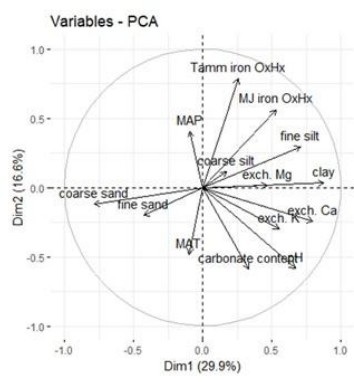

**Figure A1: Score of the 993 samples on axes 1 and 2 of the principal component analysis on 14 pedoclimatic parameters: clay, fine silt, coarse silt, fine sand and coarse sand contents; pH in water; carbonate content; mean annual temperature and mean annual precipitation; Tamm and Mehra–Jackson iron oxyhydroxide contents; exchangeable calcium, magnesium and potassium ions. Samples with an organic carbon yield between 0.7 and 1.3 are plotted in light yellow, whereas samples with an organic carbon yield $< 0.7$ or $> 1.3$ are plotted in dark red.**






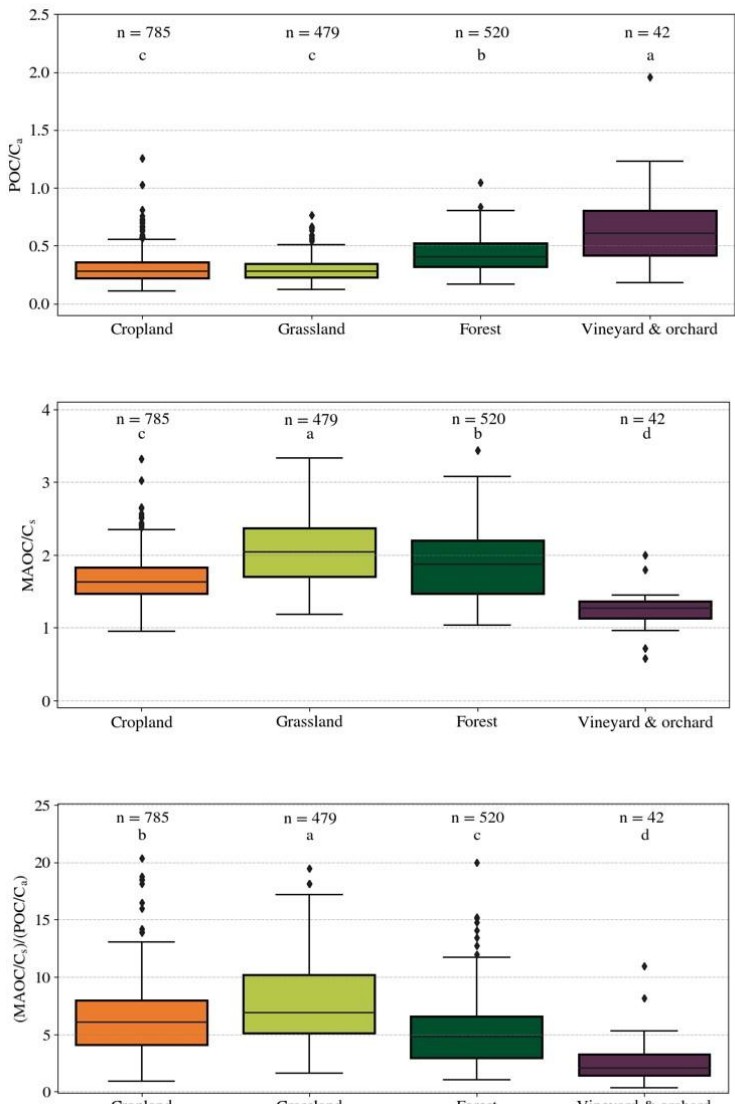

**Figure A2: Ratios of the fractions for the four major land covers. The black line in each box is the median, the lower and upper edges of the black rectangle are the respective first (Q1) and third (Q3) quartiles, and the lower and upper whiskers are the maximum between the minimum value or the first quartile minus 1.5 times the interquartile range (max [min; Q1-1,5×(Q3-Q1)]) and the minimum between the maximum or the third quartile plus 1.5 times the interquartile range (min [max; Q3+1,5×(Q3-Q1)]), respectively. Different letters indicate significant differences in the distribution of the values for the land covers according to a Kruskal–Wallis test (p < 0.05) and a pairwise Wilcoxon rank sum test (p < 0.05).**





| | $C_a$ quantity | $C_s$ quantity | TOC | MAOC quantity | POC quantity |
|---|---|---|---|---|---|
| RF train accuracy | 0.639 | 0.707 | 0.667 | 0.690 | 0.668 |
| RF test accuracy | 0.574 | 0.616 | 0.583 | 0.525 | 0.352 |
| Number of estimators | 200 | 600 | 400 | 600 | 200 |
| Maximum number of features per tree | 0.33 | 0.33 | 0.33 | 0.33 | 0.33 |
| Maximum tree depth | 4 | 4 | 4 | 4 | 4 |
| Minimum number of samples per node | 5 | 3 | 5 | 5 | 3 |
| Minimum number of samples per leaf | 3 | 3 | 3 | 5 | 3 |

**Table A1: Hyper-parameters used by the Random Forest model for TOC and the four fractions of SOC.**





| | | $C_s$ content | $C_a$ content | $C_s$ proportion | | MAOC content | POC content | POC proportion | | $C_s$/MaOC | MaOM - $C_s$ | POM/$C_a$ | $C_a$ - POM |
|---|---|---|---|---|---|---|---|---|---|---|---|---|---|
| clay | n=1,879 | 0.52 *** | 0.23 *** | 0.15 *** | n=958 | 0.39 *** | 0.18 *** | -0.15 *** | n=843 | 0.17 *** | 0.17 *** | -0.06 b | 0.22 *** |
| total silt | n=1,879 | -0.04 b | -0.12 *** | 0.17 *** | n=958 | -0.04 b | -0.18 *** | -0.37 *** | n=843 | -0.06 a | -0.05 b | -0.22 *** | -0.12 *** |
| total sand | n=1,879 | -0.28 *** | -0.04 a | -0.22 *** | n=958 | -0.20 *** | 0.03 b | 0.36 *** | n=843 | -0.06 b | -0.06 a | 0.20 *** | -0.04 b |
| TOCea | n=1,879 | 0.91 *** | 0.98 *** | -0.53 *** | n=960 | 0.96 *** | 0.86 *** | 0.32 *** | n=843 | -0.27 *** | 0.82 *** | 0.01 b | 0.90 *** |
| C/N | n=1,879 | 0.11 *** | 0.29 *** | -0.40 *** | n=959 | 0.12 *** | 0.39 *** | 0.60 *** | n=843 | -0.03 b | 0.11 ** | 0.30 *** | 0.17 *** |
| MAT 1969–1999 | n=1,879 | -0.30 *** | -0.45 *** | -0.39 *** | n=960 | -0.38 *** | -0.30 *** | -0.08 * | n=843 | 0.33 *** | -0.41 *** | 0.11 ** | -0.40 *** |
| MAP 1969–1999 | n=1,879 | 0.34 *** | 0.37 *** | -0.27 *** | n=960 | 0.37 *** | 0.29 *** | 0.14 *** | n=843 | -0.16 *** | 0.35 *** | -0.02 b | 0.38 *** |
| pH | n=1,879 | 0.15 *** | -0.20 *** | 0.55 *** | n=960 | -0.12 *** | -0.17 *** | -0.21 *** | n=843 | 0.43 *** | -0.26 *** | 0.04 b | -0.25 *** |
| InoC | n=1,879 | 0.11 *** | -0.03 b | 0.16 *** | n=958 | 0.00 b | -0.04 b | -0.05 b | n=843 | 0.14 *** | -0.04 b | -0.03 b | -0.03 b |
| Tamm iron | n=1,610 | 0.38 *** | 0.43 *** | -0.25 *** | n=823 | 0.51 *** | 0.28 *** | -0.09 * | n=714 | -0.24 *** | 0.49 *** | -0.16 *** | 0.49 *** |
| Mehra–Jackson iron | n=1,609 | 0.49 *** | 0.27 *** | 0.06 * | n=822 | 0.39 *** | 0.23 *** | -0.05 b | n=713 | 0.09 * | 0.22 *** | 0.03 b | 0.25 *** |
| CEC | n=1,879 | 0.61 *** | 0.31 *** | 0.13 *** | n=960 | 0.39 *** | 0.27 *** | -0.05 b | n=843 | 0.21 *** | 0.16 *** | 0.02 b | 0.21 *** |
| exch. Ca | n=1,879 | 0.56 *** | 0.25 *** | 0.18 *** | n=960 | 0.33 *** | 0.20 *** | -0.08 * | n=843 | 0.24 *** | 0.10 ** | 0.02 b | 0.15 *** |
| exch. Mg | n=1,879 | 0.25 *** | 0.15 *** | 0.04 a | n=960 | 0.15 *** | 0.15 *** | 0.07 * | n=843 | 0.15 *** | 0.02 b | 0.07 * | 0.07 * |
| exch. K | n=1,879 | 0.10 *** | -0.04 a | 0.18 *** | n=960 | 0.03 b | -0.08 * | -0.21 *** | n=843 | 0.15 *** | -0.06 b | -0.07 * | -0.05 b |
| exch. Al | n=1,879 | 0.11 *** | 0.39 *** | -0.45 *** | n=960 | 0.37 *** | 0.33 *** | 0.21 *** | n=843 | -0.33 *** | 0.43 *** | -0.01 b | 0.45 *** |
| exch. Fe | n=1,879 | 0.26 *** | 0.35 *** | -0.22 *** | n=960 | 0.28 *** | 0.31 *** | 0.17 *** | n=843 | -0.10 ** | 0.25 *** | 0.08 * | 0.29 *** |
| exch. Na | n=1,879 | -0.01 b | -0.02 a | 0.08 *** | n=958 | -0.02 b | 0.00 b | 0.09 ** | n=843 | 0.04 b | 0.01 b | 0.04 b | 0.02 b |
| exch. Mn | n=1,851 | -0.04 a | 0.09 *** | -0.18 *** | n=947 | -0.04 b | 0.05 b | 0.12 *** | n=832 | -0.15 *** | -0.00 b | 0.04 b | -0.01 b |

**Table A2: Spearman correlation coefficients of the Ca content, Cs content, Cs proportion, POC content, MAOC content, POC proportion, Cs/MAOC, MAOC - Cs, POC/Ca, and Ca - POC with the following pedoclimatic variables for the RMQS topsoil (0–30 cm) samples: clay, total silt, total sand, TOCea, C/N ratio, mean annual temperature (MAT) averaged over 1969–1999, mean annual precipitation (MAP) averaged over 1969–1999, pH in water, carbonate content, Tamm iron oxyhydroxides (crystalline), Mehra–Jackson iron oxyhydroxides (amorphous), cation exchange capacity (CEC), exchangeable calcium, exchangeable magnesium, exchangeable potassium, exchangeable aluminium, exchangeable iron, exchangeable sodium, and exchangeable manganese. The analysis was limited to samples meeting the required criterion for Rock-Eval® thermal analysis and/or physical fractionation. The number of samples presenting data for each calculation is indicated. Absolute values ≥ 0.3 are in bold. The asterisks and superscript letters indicate the p value: *** between 0 and 0.001, ** between 0.001 and 0.01, * between 0.01 and 0.05, a between 0.05 and 0.1, and b > 0.1. The values in grey have non significant p values.**