# Peer review of "Investigating the complementarity of thermal and physical soil organic carbon fractions"

_EGUsphere, 2024_

## Author Response (AR1)

**Point-by-point responses to R1's, R2's and Topic Editor's comments**

**Reply to RC1**

The authors warmly thank the referee for his/her supportive review! Please find below our answers (plain text) to each specific comment (italics). Additions to answers are in red. The modifications effectively applied to the manuscript appear in orange.

*A very well-written and relevant study. I have no objections to publication.*

*Minor:*

*159: The term "fPOM" can lead to confusion because fPOM has been used for several decades and in many studies as a synonym for free POM.*

**Answer:** As it only appears twice in the manuscript, there is actually no need for an abbreviation; we will write fine POM in full instead of fPOM to avoid confusion, and do the same with coarse POM for consistency.

**Edit:** We wrote fine POM and coarse POM in full.

*190: It might be worthwhile to explain why one-hot encoding was performed. It is not immediately necessary for random forests.*

**Answer:** We thank the reviewer for this important question which we will clarify in the article as well. While the theoretical RF model can handle categorical features by definition, its implementation in the scikit-learn library in Python does require encoding categorical features as they are not natively supported. We decided to use one-hot encoding as it is one of the rare feature encoders that do not establish any kind of ranking between the features (as would, for example, the ordinal or target encoding). Since our only categorical feature is the land cover, it was important to make sure that the encoding does not introduce an artificial ranking of land covers, and the small number of categories (four different land covers) does not significantly increase the number of features used.

**Edit:** We added a sentence to clarify this choice and highlight the potential ranking problem.

*342+: I have several questions regarding this matter, which may not be immediately resolvable here. It's possible that the authors possess more detailed insights and could provide further clarification. However, the importance of the publication remains unchanged, regardless of whether these questions are fully answered.*

*As described, it appears that the targeted SOC storage mechanisms of MAOC fractions and $C_S$ fractions differ. The physical fractionations aim to separate OC protected by mineral binding. As demonstrated in various studies and outlined in the script, these fractions still contain fine POC and soluble compounds.*

*However, it remains unclear to what extent the thermostability determined in Rock-Eval is linked to SOC storage mechanisms. Is it plausible that thermostability correlates solely with the recalcitrance of the treated OC? MAOC consists to a larger extent of polysaccharides and lipids. When MAOC with a potentially high MRT is treated with Rock-Eval, will it be attributed to the $C_a$ pool due to the low thermostability of polysaccharides and lipids?*

*Consequently, does this imply that the MAOC fractions will have a wider MRT range, spanning from very young (DOC/fine POM) to very old, while the Cs fraction is limited to a narrow MRT range, failing to capture either young or very old MAOC?*

**Answer:** Rock-Eval® thermal analyses give information on the chemical composition of organic matter (HI and OIre6 being proxies of H/C and O/C ratios) and its thermal stability. This information is likely related to the biogeochemical stability of SOC (Gregorich et al., 2015). Indeed, chemical composition can be linked to organic matter recalcitrance and thermal stability can give insights on OM-minerals interactions as organic molecules bound to mineral surfaces would probably evolve at higher temperature. However, we agree that to a large extent, the precise link between information given by Rock-Eval® thermal analyses and the mechanisms explaining SOM persistence in soils (guided by the return-on-energy-investment; Barré et al., 2016; Henneron et al., 2022) remains to be clearly established.

The calculation of $C_s$ proportion using the PARTYsoc model takes into account both information on chemical composition and thermal stability. The $C_s$ fraction corresponds to very stable organic matter, whether due to its chemical composition (such as PyC) or its strong interaction with minerals. The $C_s$ fraction corresponds to SOC that is not mineralized during a period exceeding a century. If it indeed does not include low-stability MAOC, it does include very old MAOC.

**Edit:** We added some precisions about the information seen by Rock-Eval® thermal analysis (chemistry + mineral interactions) in the Material and methods to clarify the potential of this technique.

References:

Barré, P., Plante, A.F., Cécillon, L. et al. The energetic and chemical signatures of persistent soil organic matter. Biogeochemistry **130**, 1–12 (2016). https://doi.org/10.1007/s10533-016-0246-0

Henneron, L., Balesdent, J., Alvarez, G. et al. Bioenergetic control of soil carbon dynamics across depth. Nat Commun 13, 7676 (2022). https://doi-org.insu.bib.cnrs.fr/10.1038/s41467-022-34951-w

Gregorich, E. G., Gillespie, A. W., Beare, M. H., Curtin, D., Sanei, H., and Yanni, S. F. Evaluating biodegradability of soil organic matter by its thermal stability and chemical composition. Soil Biol. Biochem., 91, 182–191 (2015). https://doi.org/10.1016/j.soilbio.2015.08.032

**Reply to RC2**

The authors warmly thank the referee for his/her stimulating review! Please find below our answers (plain text) to each specific comment (italics). Additions to answers are in red. The modifications effectively applied to the manuscript appear in orange.

*Delahaie et al. present results from a large dataset of physical and thermal soil fractions, and explore the correspondence between MAOC, POC, Cs, and Ca and their environmental controls. In all this is a really interesting and well-written manuscript that was a pleasure to read. I recommend publication, and include some non-critical questions/comments below for further discussion.*

*The PartySOC model was trained on long-term agronomic experiments, as also stated on line 138. Are there uncertainties in applying the model directly to grasslands and forests? Especially since it is a machine learning model. Are certain soils and vegetation types outside the ranges used to train it? Does land-use history matter? There is some discussion of this on lines 323-325, but perhaps this could be expanded.*

**Answer:** This is a good point. The major issue raised by models such as Random Forest is extrapolation: by their tree-based nature, these can't extrapolate outside values that are present in the calibration dataset. PARTYsoc was trained on long-term agronomic experiments, however, some grassland samples were also included in the training set. The TOC content of samples included in PARTYsoc's training set ranges from ca. 5 to 42 gC/kg. 85% of the RMQS samples analyzed in our study fall within this range, preventing most of the problems due to said extrapolation. Even though the model has not been trained with forest soils, the results obtained on forest samples appear realistic and consistent. Nonetheless, we agree that some discussion on this aspect has to be added in the revised draft.

We do agree that land-use history matters, and we think that PARTYsoc can reflect land-use history: our hypothesis is that the croplands with "low" $C_s$ proportions are former grasslands or forests recently converted to agriculture, while grasslands or forests with high $C_s$ proportions were likely croplands not so long ago, before being afforested or converted to grasslands. However, this hypothesis remains to be validated.

**Edit:** We added these points in the discussion at the paragraph mentioned by the referee.

*The discussion of POC/MAOC ratios on lines 334-336 is quite interesting, also in the context of this ratio being used as a proxy to assess vulnerability (e.g., as in Viscarra Rossel et al. 2019; Sanderman et al. 2021, though their fractionation schemes are slightly different than the ones here). The authors could consider extending this discussion to include implications for the broader use of this ratio.*

**Answer:** Thank you for the suggestion. Indeed, discussing the results of these articles will enrich our draft. The fractionation schemes are indeed different but it would be interesting to compare the thermal fractionation ($C_a/C_s$) and the 3 components physical fractionation schemes used in the two articles. These articles also provide correlations between POC content and pedo-climatical variables that are worth discussing with our results. Nonetheless, our results suggest that the "potential vulnerability" proposed in Viscarra Rossel et al. (2019) can be misleading in some cases, such as low TOC soils with ligneous POM.

**Edit:** We added some discussion about the specific vulnerability of the POC and $C_a$ fractions in croplands compared to grasslands and forests, where the loss rate is indeed very high (see new table A3 in supplementary), highlighting a ranking (POC > $C_a$ > MAOC > $C_s$) that is the opposite to the one for MRT, which is consistent. We also compared briefly the drivers identified for the fractions by Viscarra-Rossel et al. 2021

*The finding that the ratios of MAOC/Cs and POC/Ca differ from 1 is interesting, but these ratios themselves do not necessarily tell us how much of the Ca or Cs might come from POC or MAOC. Indeed, even if POC/Ca = 1 and MAOC/Cs = 1, there could still be many possible ways to achieve this; for instance, with most of the Ca coming from POC or MAOC (which also relates to Reviewer #1's question on interpreting thermostability). Do the authors have any ideas on how to further constrain how much Ca comes from POC vs MAOC, and similarly for Cs? This doesn't need to be resolved here, of course, but maybe the authors could discuss further insights or challenges for future work.*

**Answer:** It is difficult to constrain the composition of the $C_a$ and $C_s$ compartments. An idea could be to analyze directly POC and MAOC fractions with Rock-Eval thermal analysis, and then apply the PartySOC model to the results. However, in this case, the first limitation this review pointed out will be even more true: the PartySOC model has not been designed to be applied to fractions. Another option may be to analyze using Rock-Eval® soil samples before and after removing specifically POM (by flotation). This is indeed an interesting research avenue.

POC and MAOC are fractions that do not have a single mean residence time. The best way to know how much $C_a$ comes from the MAOC physical fraction would be to estimate the size of the compartments with a MRT of ca. 30 years and with an infinite MRT of the MAOC fraction using $\delta^{13}C$. However, this could be only achieved in chronosequences.

**Edit:** This question was added as potential future work.

*The ratio (MAOC/Cs)/(POC/Ca), as provided in Figure A2 and referenced on line 266, seems somewhat difficult to interpret and is not discussed again in the text. How do the authors think about this quantity?*

**Answer:** This ratio is indeed confusing. We thought it would help understand the size of the different compartments depending on the land use, but it seems not to be suitable for clear interpretation. We propose to remove this panel and discussion in the revised version, as it does not provide precise information we were looking for.

**Edit:** We removed the last panel of Figure A2 and the corresponding reference.

*Could the authors include bar graphs of absolute fraction quantities colored by land use in the supplement; as a Figure A3, for instance?*

**Answer:** This figure will be added to the revised version of the manuscript.

**Edit:** We added this figure in the supplement as requested, as well as a new table in the supplement to show the means for each fraction in the four land covers.

*Line 249: Could state the mean value of the Ca proportion for vineyards and orchards here for consistency.*

**Answer:** This information will be added to the revised version of the manuscript.

**Edit:** The mean value of the $C_a$ proportion for vineyards (0.35) has been added.

*Line 252: Possibly change to "As with the median values, …" instead of "As for…", otherwise reword for clarity.*

**Answer:** This change will be done in the revised version of the manuscript.

**Edit:** We changed "As for the median values […]" into "As with the median values […]".

*References noted above:*

*Sanderman et al. Soil organic carbon fractions in the Great Plains of the United States: an application of mid-infrared spectroscopy. Biogeochemistry (2021).*

*Viscarra Rossel et al. Continental-scale soil carbon composition and vulnerability modulated by regional environmental controls. Nature Geoscience (2019).*

**Reply to Topic Editor**

*The manuscript "Investigating the complementarity of thermal and physical soil organic carbon fractions" by Delahaie et al. has been reviewed by two independent reviewers, both of which have*

*highlighted its value. However, both reviewers have also highlighted the need for some improvements before the study can be published. Specifically, the uncertainties and unknowns of the approach should be discussed in more detail. Further, reviewer 1 asked for a better clarification and critical discussion on the link between Rock-Eval predicted Ca/Cs and SOC storage/formation mechanisms (or the lack thereof), while reviewer 2 asked for a similar clarification with regard to the Ca/Cs and POC/MAOC ratios. To avoid confusion on how to interpret Ca and Cs, the authors may even consider referring to Cs and Ca as conceptual pools instead of fractions.*
*I am looking forward to receiving the revised version of this manuscript.*
*Kind regards,*
*Moritz Laub*

We thank the Topic Editor for accepting our manuscript.

We addressed the requests from Referees 1 and 2 (detailed above). We specify in the introduction that thermal fractions can be seen as conceptual pools. However, $C_a$ and $C_s$ correspond to two complementary parts of a whole (the part that is stable on the scale of the century and the part that is not), which we believe corresponds to the definition of fractions.

Kind regards,

Amicie Delahaie